# A Whole Genome Sequencing-Based Epidemiological Investigation of a Pregnancy-Related Invasive Listeriosis Case in Central Italy

**DOI:** 10.3390/pathogens11060667

**Published:** 2022-06-08

**Authors:** Valeria Russini, Martina Spaziante, Bianca Maria Varcasia, Elena Lavinia Diaconu, Piermichele Paolillo, Simonetta Picone, Grazia Brunetti, Daniela Mattia, Angela De Carolis, Francesco Vairo, Teresa Bossù, Stefano Bilei, Maria Laura De Marchis

**Affiliations:** 1Food Microbiology Unit, Istituto Zooprofilattico Sperimentale del Lazio e della Toscana “M. Aleandri”, 00178 Rome, Italy; valeria.russini-esterno@izslt.it (V.R.); biancamaria.varcasia@izslt.it (B.M.V.); teresa.bossu@izslt.it (T.B.); stefano.bilei@izslt.it (S.B.); 2Regional Service Surveillance and Control for Infectious Diseases (SERESMI), National Institute for Infectious Diseases “Lazzaro Spallanzani” IRCCS, 00149 Rome, Italy; martina.spaziante@inmi.it (M.S.); francesco.vairo@inmi.it (F.V.); 3Department of General Diagnostics, Istituto Zooprofilattico Sperimentale del Lazio e della Toscana “M. Aleandri”, 00178 Rome, Italy; elena.diaconu-esterno@izslt.it; 4UO Neonatologia, Patologia Neonatale e Terapia Intensiva Neonatale (TIN), Policlinico Casilino General Hospital, 00169 Rome, Italy; pmpaolillo.polcas@eurosanita.it (P.P.); simpico@libero.it (S.P.); 5Pathology-Microbiology Laboratory, Policlinico Casilino General Hospital, 00169 Rome, Italy; gbrunetti.polcas@eurosanita.it; 6Dipartimento di Prevenzione, Servizio Veterinario Area B—Igiene Alimenti di Origine Animale (SIOA), ASL Roma 6, 00072 Rome, Italy; daniela.mattia@aslroma6.it; 7Dipartimento di Prevenzione, Servizio di Igiene degli Alimenti e della Nutrizione (SIAN), ASL Roma 6, 00044 Rome, Italy; angela.decarolis@aslroma6.it

**Keywords:** listeriosis, *Listeria monocytogenes*, pregnancy, maternal-fetal transmission, outbreak, next generation sequencing, WGS, foodborne

## Abstract

Listeriosis is currently the fifth most common foodborne disease in Europe. Most cases are sporadic; however, outbreaks have also been reported. Compared to other foodborne infections, listeriosis has a modest incidence but can cause life-threatening complications, especially in elderly or immunocompromised people and pregnant women. In the latter case, the pathology can be the cause of premature birth or spontaneous abortion, especially if the fetus is affected during the first months of gestation. The causative agent of listeriosis, *Listeria monocytogenes*, is characterized by the innate ability to survive in the environment and in food, even in adverse conditions and for long periods. Ready-to-eat food represents the category most at risk for contracting listeriosis. This study presents the result of an investigation carried out on a case of maternal-fetal transmission of listeriosis which occurred in 2020 in central Italy and which was linked, with a retrospective approach, to other cases residing in the same city of the pregnant woman. Thanks to the use of next-generation sequencing methodologies, it was possible to identify an outbreak of infection, linked to the consumption of ready-to-eat sliced products sold in a supermarket in the investigated city.

## 1. Introduction

The Gram-positive *Listeria monocytogenes* is a facultative intracellular bacterium responsible for the food-borne disease listeriosis [1,2,3], an infection predominantly transmitted through the consumption of contaminated food.

Listeriosis has a wide spectrum of infections, from severe invasive listeriosis, non-invasive febrile gastroenteritis [4] to asymptomatic forms [5]. As matter of fact, *L. monocytogenes* occurs in stool of asymptomatic carriers probably due to a less diverse microbiota compared to non-carriers [5]. The microbiota diversity is also involved in colonization resistance against other enteropathogens [6]. Invasive listeriosis mostly occurs in immunocompromised individuals and manifests itself as sepsis, meningitis, endocarditis, encephalitis, septicemia and brain infection [7,8].

The incidence of infection ranged from 0.80 cases per 100,000 in North America in 1990 to 0.42 cases per 100,000 in Europe in 2020, mostly presenting as a sporadic disease, even if also food-borne outbreaks have been described [9,10,11]. Although a weak decrease in reported cases has been registered during 2020, probably due to the effect of the COVID-19 pandemic, the overall trend for listeriosis in 2016–2020 has shown a substantial stability. Listeriosis has the highest proportion of hospitalized cases of all zoonoses under European surveillance. The overall EU case fatality in 2020 was 13.0%, slightly decreased compared with 2019 and 2018 (17.6% and 13.6%, respectively) [10].

Nosocomial outbreaks of listeriosis have been also reported, especially among newborns and immunocompromised patients [12,13,14,15]. Listeriosis is more common in pregnancy; globally, 16–27% of *Listeria* infections occur in pregnant women [16,17,18] and maternal, fetal or neonatal origin is now estimated in about 4–10/100,000 pregnant women/year in Europe and North America [19]. During pregnancy cellular immunity may be suppressed, due to the increased progesterone levels [20,21], making pregnant women particularly susceptible to infections due to intracellular microorganisms such as *L. monocytogenes* [22,23]. The maternal-fetal cell-to-cell transmission is frequent since *L. monocytogenes* shows both uterus and placenta tropism [22,24].

Sporadic cases of perinatal listeriosis, as well as epidemic cases, have been reported with a prevalence varying between 8.6 and 17.4/100,000 of live births, with a fatality rate as high as 20% [25,26,27]. The clinical features of listeriosis during pregnancy include mild flu-like symptoms, myalgias, arthralgias, and nonspecific gastrointestinal symptoms, while about 30% of the women are asymptomatic [28,29]. Listeriosis during pregnancy carries a poorer prognosis for fetuses who are affected at early gestations [30], commonly resulting in spontaneous abortion or stillbirth [31,32]. Preterm birth is common [33], and the greatest risk of mortality and morbidity is for those infants born at the earliest gestational ages [34].

In Italy, listeriosis is a disease subject to mandatory notification, as required by the Ministerial Decree of 15 December 1990, which places it among the relevant diseases with high frequency or subject to control interventions. Starting from 2010, listeriosis has been part of the Enter-Net Italia surveillance network [35]. Subsequently, according to the “0008252-13/03/2017 DGPRE for listeriosis surveillance and prevention” circular note of the Italian Ministry of Health, the role of the National Focal Point for the “Foodborne and Waterborne diseases and zoonosis” ECDC program and *L. monocytogenes* Operational Contact Point, has been assigned to the Istituto Superiore di Sanità (ISS, Rome, Italy).

In Italy, listeriosis incidence rates significantly increased from 2014, reaching a value of 0.29 per 100,000 population in 2018 but still remaining lower when compared to the EU mean incidence rate of 0.47 per 100,000 in the same year [36]. An Italian study, focusing on invasive listeriosis cases in the Lombardy region over a ten-year period, reported that about 6.6% of the confirmed cases were pregnancy-related (2005–2014) [37]. Furthermore, the frequency of Italian pregnancy-related listeriosis was lower compared with the rates in other countries (Spain 7%, Belgium 10%, UK 12%, USA 16.9%, France 18%; Israel 35%) [18,37,38,39,40,41,42].

Epidemiological links and sources of contamination are rarely demonstrated, probably due to the high frequency of asymptomatic clinical presentation and to long-term incubation of *L. monocytogenes*, which can be between two and ten weeks [24,43,44].

Herein is presented a case of a pregnancy-related invasive listeriosis occurring in central Italy at the beginning of 2020. Following the epidemiological investigation, a probable source of contamination was found in a local supermarket attended by the patient. A strong epidemiological link was demonstrated, partially supported by whole genome sequencing analysis of human, food and environmental *L. monocytogenes* isolates. With a retrospective approach, the WGS analysis allowed us to identify a small outbreak that occurred in the same city as the pregnancy-related listeriosis.

## 2. Results

### 2.1. Case Presentation

On 9 January 2020, a 28-year-old pregnant woman, residing in a city near Rome and without a significant medical history, was admitted in the 31st week of gestation to the Obstetrics and Gynecology Department of Casilino Hospital in Rome, due to premature rupture of membranes (PROM) and gastrointestinal symptoms. At the admission, neutrophilic leukocytosis was documented (33,000 white blood cells/mm^3^ with 28,080/mm^3^ neutrophils), along with an increase in C-reactive protein (CRP) (90.1 mg/L). Blood cultures, serum *Listeria* IgM, screening for TORCH infections and group B *Streptococcus* analysis tested negative whereas no cerebrospinal fluid (CSF) examination was performed at that time. Ampicillin 2 gm intravenous (IV) q6h was promptly prescribed; the following day a cesarean section was performed and a 1540 gr female afebrile baby was born. Apgar at one minute was 7, than 8 at five minutes. The baby presented respiratory distress and she was diagnosed with hyaline membrane disease (HMD). Blood tests showed a noticeable increase in inflammatory indices (procalcitonin and CRP measured 0.774 mg/mL and 19.8 mg/L respectively), and the blood samples culture, collected few hours after the birth, documented the growth of *L. monocytogenes*. Antibiotic therapy with intravenous (IV) weight-adjusted ampicillin and gentamicin was promptly initiated. Lumbar puncture was also performed; cerebrospinal Fluid (CSF) was sterile, opalescent, with 100 white blood cells (WBC), mostly polymorphonuclear leukocytes, with normal glucose and protein levels.

She was transferred to the Neonatal Intensive Care Unit where she underwent to invasive ventilation for 24 h, then to non-invasive positive-pressure ventilation for further 48 h; negativization of blood cultures and normalization of procalcitonin and CRP occurred. The baby was discharged 30 days later in good clinical conditions. After the notification of the listeriosis case, an investigation was conducted by the local health authority, which administrated an epidemiological questionnaire to the mother in order to assess any potential food exposures during the 30 days before illness onset.

The mother declared that she had consumed mainly food purchased in two local supermarkets, one local butcher shop and one fish market. In particular, she reported the purchase of fresh horse meat in the butcher shop, cold cuts at the deli counter as mortadella, baked ham and salami, ready-to-eat bagged salad, other fresh vegetables and fresh fruits (apples and tangerines), dairy products as mozzarella and parmigiano in the two local supermarkets, and smoked salmon and tuna in the fish market. She also claimed to have eaten once at a local restaurant and she denied any travel abroad during the past month.

According to the mother’s declarations, the local health authority’s attention was focused on the fresh horse meat purchased in the local butchery and cold cuts at the deli counter purchased in one of the two reported supermarkets. During the different inspections, food samples and environmental samples were collected to assess the presence of *L. monocytogenes*.

The patients’ clinical course and the events related to epidemiological investigation are summarized in Figure 1.

### 2.2. Retrospective Epidemiological Investigation

After the notification of the neonatal listeriosis, additional inquiries were carried out to explore the occurrence of further listeriosis cases in the same city of the woman and her daughter (Case A). In the case series of samples collected by our regional center, three strains of *L. monocytogenes* isolated from clinical cases were found (Case B, C and D) residing in the same city, occurring from May 2019 to May 2020 and all belonging to the same serotype (see below). Where possible, patients were contacted by the competent authority to investigate any epidemiological links.

The first case (B) occurred nine months before the above-mentioned episode and involved a patient admitted to the emergency department. A lumbar puncture was performed and a diagnosis of *L. monocytogenes* meningitis was formulated. CSF cultures reported the growth of *L. monocytogenes*; antimicrobial susceptibility testing documented ampicillin, erythromycin and meropenem sensibility (MICs determined by E-test: 0.25 µg/mL, 0.38 µg/mL and 0.094 µg/mL respectively) and resistance to trimethoprim-sulfamethoxazole (0.094 µg/mL).

The second case (C) occurred three months later and a patient was hospitalized in April 2020. Blood cultures performed at the time of admission documented the growth of *L. monocytogenes*. Unfortunately, no data were available on the antimicrobial susceptibility profile.

The third case (D) occurred four months after the above-described neonatal listeriosis case, in May 2020. Seven days after the admission, due to the occurrence of fever, the patient underwent two blood cultures and *L. monocytogenes* was isolated from both samples. The strain proved to be susceptible to ampicillin (≤0.25 mg/L), erythromycin (≤0.12 mg/L), meropenem (0.19 mg/L), penicillin (0.5 mg/L) and trimethoprim-sulfamethoxazole (0.012 mg/L).

From epidemiological investigation and questionnaire collection it emerged that at least one patient (case B) confirmed their attendance at the same supermarket of the concerned neonatal listeriosis case. No further information could be obtained from the other patients.

### 2.3. Bacterial Identification and Serotyping

Cervicovaginal swabs of the mother performed after identification of *Listeria* in the newborn were negative. Unfortunately, probably due to antibiotic therapy, it was not possible to isolate the bacterium responsible for the disease, and no more investigation could be carried out on this strain. Clinical information about the first diagnosis of listeriosis and the therapy carried out are untraceable. Blood cultures drawn by the newborn reported the growth of *L. monocytogenes*, passed from the mother, presumably by vertical uterus-placenta cell-to-cell transmission, before the delivery. In this case it was possible to isolate the strain for further analysis.

For all the clinical strains antimicrobial susceptibility testing (AST) was performed as minimum inhibitory concentration (MIC). All the strains were susceptible to ampicillin (all the strains: ≤1 mg/L), meropenem (Case B and D: 0.25 mg/L; Case A and C: 0.12 mg/L) and erythromycin (all the strains: ≤0.25 mg/L)

The food samples of horse meat from the butcher shop were tested as negative for *L. monocytogenes*, meanwhile the mortadella samples from the first inspected supermarket were tested as positive, with a colony-forming unit less than 40 UFC/g. Subsequently, environmental samples were collected to identify the extent of contamination. Samples from the two meat-slicers (E1–3, G1–2) present in store were tested as positive for *L. monocytogenes*, as well as the cold food counter where the cold cuts were exposed to the public (H1–2). The worktable where the staff handled the meat (F), by contrast, was tested as negative for the contamination. All the positive strains were isolated for further analysis.

The isolates collected were assigned to the serotype 1/2b. Sequencing data confirmed serogroup IIb for the isolates and MLST analyses indicated that they all belonged to Sequence Type (ST) 5, Clonal Complex (CC) 5, lineage I.

### 2.4. In Silico Analysis

All the isolates considered in the study, four human cases (A, B, C, D), the environmental samples (E1–3, G1–2, H1–2) and food samples (M1–2), were found to bring the same genes for antibiotic resistance. The identified genes were those of resistance to the lincomycin (*lin*) with a new allele not described before, quinolone (*norB*), sulfonamides (*sul*), fosfomycin (*fosX*) antibiotics and cationic peptides that disrupt the cell membrane, including defensins (*mprF*). The virulence features were assessed and all the isolates shared the same virulence genes with the same alleles (Table 1). The isolates presented the *inlA*, *inlB* and *actA* genes that played a crucial role in the virulence and pathogenesis of *L. monocytogenes*. The presence of metal, detergent resistance, stress Survival Islands and *Listeria* Genomic Islands were the same for all the strains and were reported in Appendix A.

The cgMLST profile was determined by using two different softwares: cgMLSTFinder (v1.1) and chewBBACA (v2.8.5), described in *Material and methods* section. In the first case the analysis identified two clusters which differed by 8 alleles (Figure 2A). The first cluster was composed by human isolates from case A and case D and the second one was composed by environmental samples and human isolates from cases B and C. No differences were detected inside each cluster. Using chewBBACA, the same two clusters were inferred (Figure 2B). The human isolate from case A had 9–11 allelic differences with food and environmental samples, and showed one allelic difference with case D. The second cluster was composed of food and environmental samples, differing from each other by 2–3 alleles, and cases B and C that differed by 1–3 alleles each from environmental strains. The Minimum Spanning Tree showed the schematic differences among the isolates (Figure 2).

Further investigation was conducted by searching for ST5 *L. monocytogenes* isolates from food samples among those received by our laboratory during routine activities. Food samples were found in the records concerning chicken meat (Food Sample 1), pork meat (2), smoked swordfish (3) and salmon (4). Representative strains were used to compare food isolates with the investigated cluster, following the workflow described in materials and methods. Results showed that there was no relation with the main cluster in the core genome, with 58–106 allelic distance calculated with cgMLSTFinder and 66–115 allelic distance with chewBBACA (Appendix A).

The BIGSdb-Pasteur, the genomic-based strain taxonomy and nomenclature platform of Institute Pasteur, assigned the same profile for all food, environmental and clinical samples: CT7923. Despite the higher number of allele differences found with the other tools, according to the data available to the platform, all the samples belong to the same CT (cgMLST type) and therefore probably to the same outbreak event.

The SNPs’ analysis confirmed the tight relationship of the strains considered in this investigation. The *L. monocytogenes* strain from mortadella (M) had two SNPs (in whole and core genome) differences with the environmental isolates. The human isolates from cases A and D differed 16–19 SNPs in whole genome (10–13 in core genome) from food and environmental isolates, and human strains from cases B and C. The SNPs’ matrix-based heatmap for whole genome and core genome are reported in Figure 3 and Appendix A.

## 3. Discussion

Herein is described a case of pregnancy-related invasive listeriosis occurring in central Italy, where the infection proved to be linked to contaminated food products and vertically transmitted from mother to baby.

The mother’s infection manifested as asymptomatic form [5], with only gastro-intestinal symptoms probably confused with labor contractions.

Since it was not possible to isolate the strain of *L. monocytogenes* from the mother, the effective differences between the mother’s and the child’s strains remain unknown, even if a previous study suggested that in such cases no differences occurred, given the vertical transmission of the pathogen [45].

After a thorough completion of the epidemiological questionnaire by the mother, a strong epidemiological link with the mortadella sold in the supermarket was found, despite both the analyses of cgMLST showing allelic differences higher than the generally accepted limit to consider its inclusion in the same cluster. *L. monocytogenes* strains within seven allelic differences are considered to be linked to the same outbreak and are assigned to a unique cgMLST type (CT) [46], although this cutoff is debatable [45].

The use of different bioinformatics tools of analysis showed slightly different results. This could be due to the Prodigal gene prediction program, integrated in the chewBBACA pipeline, whose function is based on the identification of translation initiation sites [47] and, differently from cgMLSTFinder, is able to recognize new alleles when alternative start codons exist on the analyzed gene (https://github.com/B-UMMI/chewBBACA/issues (accessed on 11 February 2022).

Even if chewBBACA found 9–11 allelic differences between the newborn isolate and the food and environmental samples, thus excluding the possibility of tracing the source of contamination, the cgMLSTFinder results, on the other hand, could put it into question, finding only eight differences, at the limit of the above-mentioned threshold value. Moreover, the Institute Pasteur, whose bioinformatics analysis platform is based on chewBBACA software and which collects and characterizes isolates of *L. monocytogenes* all over the world, assigned the same CT7923 to all the strains involved in this investigation, suggesting a unique outbreak event. The overall view of their database suggests that the existence of other isolates, which act as a connection between groups, confirmed the existence of a single cluster.

Other considerations may support this hypothesis. First, the sequence type 5 (ST5) of the strains involved in this epidemiological investigation is very rare in human cases in Italy and almost limited to Lazio region (source: ISS IRIDA-ARIES database [48,49] accessed on 22 February 2022). Moreover, based on the sequence data of *L. monocytogenes* strains isolated from food samples during routine official control activities in the Lazio region by our laboratory, it seems to involve mainly pork meat (data not shown). These peculiar features, together with a strong epidemiological link found between the specific category of food, the geographical area and the habits of the mother of patient A during pregnancy and patient B, could support the hypothesis that all the reported episodes belong to the same outbreak event.

The differences found among samples with SNPs and cgMLST analysis could be due to the different time of isolation. Considering that inspections started one month later than the symptoms’ onset in the mother, and given the highly variable incubation period of listeriosis (17–67 days) [50], the estimated exposition time would have been dated up to three and a half months before the positive sample collection.

In a recent work by Luo and colleagues in 2019, [45], it was found one SNP of difference in core genome between mother and child strains isolated 13 days apart, despite the mutation rate of *L. monocytogenes* core genome is estimated at 2.6 × 10^−7^ substitutions per site per year (one substitution every 2.5 years) [46]. Thus, a much higher mutation rate, at least in clinical isolates or in pregnancy-associated listeriosis, may be suspected. As a matter of fact, the clinical isolates could evolve faster than environmental and food strains. Environmental temperature may be extremely variable, but never exceeds 20–22 °C. Food storage temperature is controlled between −20 °C and room temperature. These conditions, although allowing growth of *L. monocytogenes*, significantly increase the generation time of the organism compared to the optimal growth temperature (typically 30–37 °C) [51]. Furthermore, treatment of surfaces with cleansing and sanitizer could represent a stress factor for *L. monocytogenes* growth [52,53].

In addition to differences in generation time and variations in population sizes, the direction (i.e., negative or positive) and strength of natural selection could also contribute to differences in the rate of changes per site per year [54]. Given this assumption, the mutation rate could be higher than that estimated, which should explain the longer distance between clinical strains A and D and the suspected food and environmental strains.

Another hypothesis could be the occurrence of the analysis on a different portion or batch of the mortadella originally consumed, with a different strain than the original source of contamination, which could since have been completely sold by the supermarket and not available during the inspection. Alternatively, since it was not possible to isolate the maternal strain, it could have happened that the mother was originally infected with more than one strain of *L. monocytogenes*, not all transmitted or found to the newborn [55,56,57].

The analysis, through the use of WGS, showed that the three retrospectively identified human cases were closely linked with the newborn case A and with food and environmental samples. In Figure 4a schematic view of the events and relationships was reported. In particular, cases B and C were strictly linked to the supermarket and mortadella. Case B occurred nine months before the inspections and sampling at supermarket and differs 1–3 SNPs from food and environment (2–3 SNPs in core genome). The patient confirmed that he used to attend the concerned supermarket. Case C occurred more than two months after the sampling date, and present two SNPs with food strains and four SNPs with environmental strains two and three SNPs in core genome). Case D occurred more than three months after the supermarket sampling, presented the same amount of divergence with food and environment strains as case A, differing three SNPs in whole genome (one SNP in core genome). Therefore, the molecular investigation suggested that the source of contamination, in any case, had been active for a long period both before and after the official inspection by local heath authority.

After the correct sanitation of premises, sampling performed in self-control and by the local health authority tested negative. Consequently, the deli counter activity was reopened, according to current legislation. The contamination of mortadella was most likely secondary, since the sampled product was already open and no data relating to an intact package of the same batch were available.

Based on that, it could be concluded that the source of contamination in the investigated area was eliminated, and that the actual food source of contamination remains to be investigated. Beyond the strictly sanitary purposes and given the molecular results that are not completely conclusive, further investigations on other probable contamination sources would probably have been useful, in order to assess if the contaminated food found in the inspections was the only one among the products consumed by the mother.

Since the used methodologies have highlighted tools-related results biases, the use of more than one method for analysing data related to an outbreak would be recommended, to avoid errors of underestimation or overestimation of the diversity between the involved strains. Our results also show the possibility that the cut-off of seven allelic differences represents a criterion that is sometimes too stringent for the definition of an epidemic cluster of *L. monocytogenes* and also suggest other factors, such as the frequency of the investigated sequence type and the presence of other strong epidemiological links.

Herein is described the first outbreak linked to the ST5 in Italy, which also involved a pregnancy-related case. This specific ST, based on national databases, is not common in Italy and seems to be limited in central Italy, in particular the Lazio region (source ISS database). Concerning Europe, sporadic occurrences of ST5 have been reported in literature. Several cases, mostly linked to maternal-fetal listeriosis, were reported during 1992–2003 in France [58]. An investigation into an Austrian cheese processing facility [59] detected the predominance of ST5 during a two-year monitoring of the processing environment (2010–2012). A study carried out in Poland between 2014 and 2017 of different kinds of ready-to-eat food of animal origin and food processing environments showed a high prevalence of ST5, found in 22.9% of samples, all derived from pork meat sausages and their processing plants [60]. Outside Europe, an outbreak was reported in the USA, linked to two ice-cream facilities. Clinically isolated, food and environmental samples all showed the ST5 [61]. Furthermore, this sequence type was found to be very frequent in Asia, especially in the central area of China [62], and was identified in clinical and pregnancy-related cases in Beijing [63]. The importance of reporting outbreaks linked to ST5, in particular when pregnancy-related, is reinforced by the sporadic occurrence of this ST worldwide.

Foods purchased from store delicatessen counters are a risk factor for sporadic listeriosis. Turkish delicatessen meat, pasteurized butter, and hot dogs have all caused outbreaks of *L. monocytogenes* in the past [18]. This experience shows that continued education about dietary precautions for all pregnant women is required to further reduce the occurrence of listeriosis. In addition, clinicians should consider the possibility of listeriosis whenever a pregnant woman presents with non-specific or flu-like symptoms because early diagnosis and appropriate antibiotic treatment can be lifesaving and significantly improve pregnancy outcome [18].

Finally, this study highlights the importance of a constant surveillance activity of foodborne diseases carried out by health authorities together with clinicians and laboratorians, from notification to investigation. The major aim is to identify and stop the source of contamination as soon as possible, to avoid the extension of a foodborne outbreak.

## 4. Materials and Methods

### 4.1. Food and Environmental Sampling

After the assessment of the epidemiological questionnaire completed by the mother, the Veterinary Service of Hygiene for Foods of Animal Origin (SIAOA) of the local health authority conducted sampling at the butcher shop and the deli counter of the attended supermarket involved. Sampling of horse meat and mortadella was performed according to the Reg. (CE) 2073/2005. Briefly, by means of sterile scalpels four aliquots were aseptically collected, each consisting of five sample units of 120 g, and stored into sterile bags at +4 °C. Environmental sampling was performed by means of sterile swabs and containers. In particular, two meat slicers (E, G), the worktable for handling cold cuts (F), and a refrigerated counter for exposed and store-cold cuts (H) were sampled, each with more than one swab. All the collected samples and times of sampling are summarized in Table 2.

### 4.2. Microbiological Methods for Bacterial Identification and Serotyping

A blood sample of the newborn was collected, inoculated in Bactec Plus aerobic and anaerobic broth, and incubated in the automated culture system, Bactec 9420 (Becton Dickinson, Inc., Sparks, MD, USA). The positivity was assessed upon direct observation under the optical microscope of motile bacilli and stained with Gram dye. An aliquot of the sample was plated using WASP^®^ automated specimen processing (Copan srl, Brescia, Italy) on Columbia Agar, Columbia CNA Agar, Mac Conkey II Agar, Mannitol Salt Agar, BBL CHROMagar Candida Medium and Chocolate Agar (Becton Dickinson, Inc., Sparks, MD, USA) and incubated at 37 °C for 24 h. Therefore, the multiplex PCR was performed on an aliquot of the positive sample (FilmArray^®^ bioMérieux SA, Marcy-l’Etoile, France). This technique resulted in the detection of 33 pathogens and 10 antibiotic resistance genes one hour after blood culture positivization. The identification of *L. monocytogenes* was also confirmed, with a score ≥2, by Maldi-TOF mass spectrometry (Bruker Daltonik GmbH, Bremen, Germany) performed on plate-cultured colonies after 24 h of incubation.

All the isolated clinical strains were sent to the Regional Reference Centre for Pathogenic Enterobacteria (CREP) at the Food Microbiology Unit of IZSLT—central division of Rome.

The food and environmental samples collected by LHA were sent to Food Microbiology Unit laboratories of IZSLT. The detection of *L. monocytogenes* was performed by Real-time PCR using iQ-Check *Listeria monocytogenes* II PCR Detection Kit (Bio-Rad Laboratories, Inc., Segrate, Italy). Positive samples were cultured according to UNI EN ISO 11290-1:2017, using ALOA medium and LSM as the second selective medium. *L. monocytogenes* identification was confirmed by VITEK^®^ 2 Compact System (bioMérieux SA, Marcy-l’Etoile, France). The biochemical culture confirmation was carried out performing the beta-hemolysis test, the L-rhamnose test, and the D-xylose test.

All the isolated strains were serotyped by using the Mast assure antiserum *Listeria* ‘H’ and ‘O’ (Mast Group) according to the manufacturer’s instructions at CREP laboratory of IZLST. Antimicrobial susceptibility testing (AST) for all the clinical strains was performed as minimum inhibitory concentration (MIC) determination by broth microdilution (Trek Diagnostic Systems, Westlake, OH, USA), according to the European Committee on Antimicrobial Susceptibility Testing (EUCAST; http://www.eucast.org (accessed on 28 April 2022), using epidemiological cut-offs. The following drugs were tested: Meropenem, Ampicillin, Erythromycin. *Streptococcus pneumoniae* ATCC 49619 was used as a quality control strain.

### 4.3. Whole Genome Sequencing and In Silico Analysis

For whole genome sequencing analysis, one colony was selected for each food sample unit tested positive and one colony for each swab used for environmental sampling (Table 2). Genomic DNA was extracted with the automatic extraction system QIAsymphony (Qiagen, Hilden, Germany). Libraries were prepared using Nextera XT DNA Library Prep and pair-end (2 × 300 bp) run with a MiSeq sequencer (Illumina, CA, USA).

Raw reads quality was assessed with Fast QC (v0.11.5) [64] and low-quality reads and adapters were trimmed using Trimmomatic (v0.39) [65] using the following quality filter: minimum quality of Q30, a window size of 10 with Q20 as average quality, and a minimum length read of 50 bp. The high quality reads were de novo assembled into contigs using SPAdes (v3.13.0) with the careful option on [66], draft assemblies were improved using Pilon (v1.23) [67] and contigs shorter than 500 bp were removed [68]. The assembly quality was assessed with QUAST (v5.0.2) [69]. The serogroup in silico was deduced by evaluating the composition of selected loci [70]: lmo0737, lmo1118, ORF2110, ORF2819, *prs* (lmo0199) with BLAST (v2.11.0) [71]. In silico subtyping was performed with MLST (v2.11) [72] that used the seven housekeeping genes (*abcZ*, *blgA*, *cat*, *dapE*, *dat*, *ldh*, and *lhkA*) retrieved from the BIGSdb-Lm [73].

The cgMLST analyses were performed using two pipelines. The first one was cgMLSTFinder (v1.1 developed by the Center for Genomic Epidemiology), that runs KMA [74] against the *Listeria* cgMLST database [75] to create core genome allele profiles from raw sequence data (1748 highly conserved core genes). This software assesses the presence of only already known alleles in genomes and considers the possible new one as not found.

The second adopted pipeline was chewBBACA (v2.8.5) [76] based on the same *Listeria* cgMLST database [75]. This software uses the tool Prodigal [47] to search for already known alleles, but also assigns new alleles if they are not identical. The newly identified cgMLST profiles of the tested samples were added at the Pasteur Institute BIGSdb-Lm. Minimum spanning trees of both results were generated using the MSTreeV2 algorithm in GrapeTree (v1.5.0) software [77].

The SNPs and core genome SNPs’ analysis was performed with the pipeline CSI phylogeny (v1.4) accessible from the Center for Genomic Epidemiology [78], starting from trimmed reads and from the core genome found by chewBBACA. The SNPs were filtered according to parameters: a minimum distance of 10 bp between each SNP, a minimum of 10× depth and 10% of the breadth coverage, the mapping quality was above 30, the SNP quality was higher than 25. The SNP matrix was visualized in a heatmap using ClustVis web tool [79].

Identification of acquired antibiotic resistance genes and genes encoding for persistence (metal, detergent resistance, stress Survival Islands and *Listeria* Genomic Islands) was assessed starting from the assemblies using BLAST against genes database available on the BIGSdb-Lm platform (accessed on 13 January 22). The same method was used to assess the presence for virulence and antimicrobial resistance genes of each strain.

## Figures and Tables

**Figure 1 pathogens-11-00667-f001:**
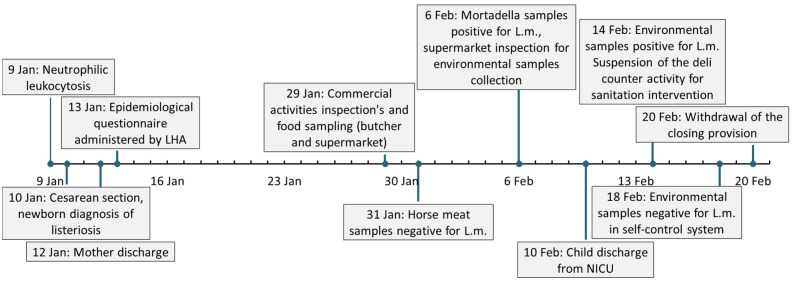
Timeline of pregnancy-related listeriosis clinical events. LHA: Local health authority. L.m.: *Listeria monocytogenes*. NICU: neonatal intensive care unit.

**Figure 2 pathogens-11-00667-f002:**
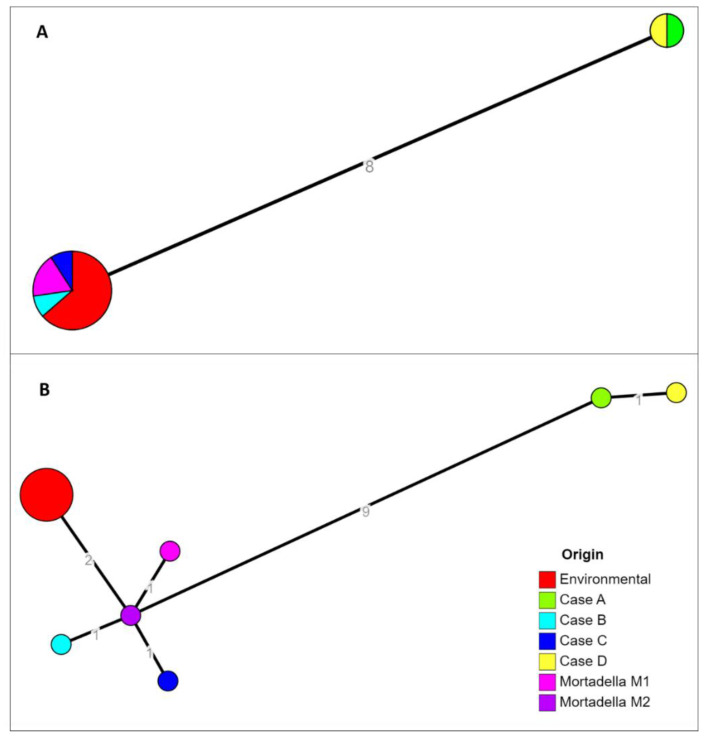
Results of cgMLST characterizations assessed with the two softwares cgMLSTFinder (**A**) and chewBBACA (**B**). Allelic distances were indicated on the network branches.

**Figure 3 pathogens-11-00667-f003:**
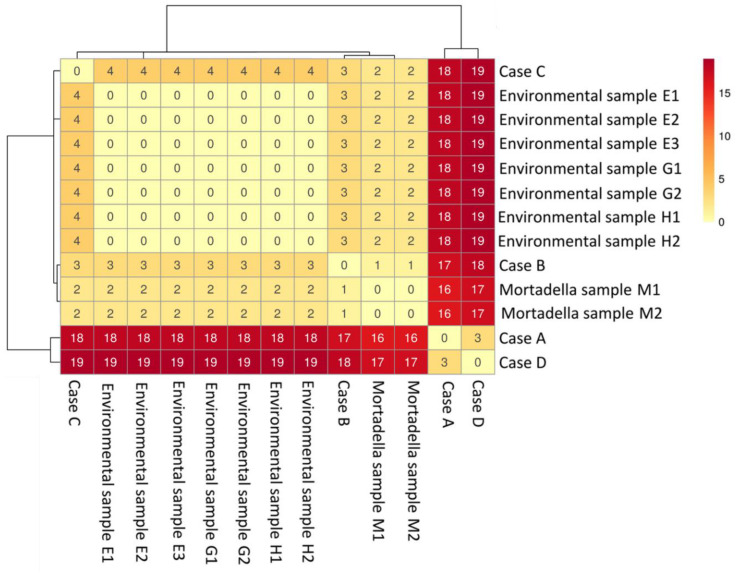
SNPs’ matrix-based heatmap showing the variation in whole genome SNPs between strains considered in this study, using trimmed reads and quality filters.

**Figure 4 pathogens-11-00667-f004:**
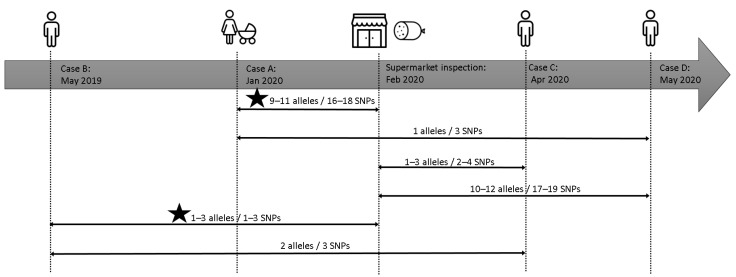
Timeline of the events described in the retrospective investigations. The lines connect and shown the genomic relationship between isolates. The star indicates the existence of a strong epidemiological link.

**Table 1 pathogens-11-00667-t001:** Complete list of virulence genes found in the analysed strains, in brackets the corresponding lmo code according to BIGSdb-Lm is indicated. For each gene the corresponding allele was indicated (BIGSdb-Lm).

Virulence Gene	Allele	Virulence Gene	Allele
*actA* (lmo0204)	120	*lisK* (lmo1378)	13
*agrA* (lmo0051)	6	*lisR* (lmo1377)	3
*agrC* (lmo0050)	12 (1 mismatch) *	lmo0333 (inlI)	17
*ami* (lmo2558)	127	lmo1280 (codY)	3
*aut* (lmo1076)	6	lmo2470 (inlP)	9
*bsh* (lmo2067)	33	lmo2491 (pdeE)	2
*cheA* (lmo0692)	13	*lntA* (lmo0438)	48
*cheY* (lmo0691)	3	*lpeA* (lmo1847)	12
*cwhA* (lmo0582; iap)	15	*lplA1* (lmo0931)	13
*dltA* (lmo0974)	13	*lspA* (lmo1844)	6
*fbpA* (lmo1829)	15	*mdrM* (lmo1617)	3
*fur* (lmo1956)	4	*mpl* (lmo0203)	16
*gtcA* (lmo2549)	7	*oatA* (lmo1291)	11
*hly* (lmo0202)	9	*oppA* (lmo2196)	8
*hpt* (lmo0838)	14	*pdgA* (lmo0415)	5
*inlA* (lmo0433)	12	*plcA* (lmo0201)	17
*inlB* (lmo0434)	15	*plcB* (lmo0205)	6
*inlC* (lmo1786)	13	*prfA* (lmo0200)	7
*inlC2* (LMON_RS01340)	6	*prsA2* (lmo2219)	5
*inlD* (LMON_RS01345)	6	*purQ* (lmo1769)	17
*inlE* (lmo0264)	4	*srtA* (lmo0929)	4
*inlF* (lmo0409)	10	*srtB* (lmo2181)	8
*inlH* (lmo0263)	12	*stp* (lmo1821)	5
*inlJ* (lmo2821)	172	*svpA* (lmo2185)	10
*inlK* (lmo1290)	14	*tagB* (lmo1088)	8
*lap* (lmo1634)	13	*vip* (lmo0320)	50
*lapB* (lmo1666)	11	*virR* (lmo1745)	4
*lgt* (lmo2482)	6	*virS* (lmo1741)	6

* New allele found.

**Table 2 pathogens-11-00667-t002:** Food and environmental samples collected for the epidemiological investigation.

ID Code	Sequenced Isolates	Food Sampling or Sampling Site	Sampling Date
L	Negative	Horse meat	29 January 2020
M	M1, M2	Mortadella	5 February 2020
E	E1, E2, E3	First meat slicer	6 February 2020
F	Negative	Worktable	6 February 2020
G	G1, G2	Second meat slicer	6 February 2020
H	H1, H2	Cold food counter	6 February 2020

## Data Availability

The authors confirm that all the Appendix A, code and protocols have been provided within the article or through Appendix A. Raw reads can be found in Sequence Read Archive (SRA) at GenBank database (NCBI) under the BioProject PRJNA814492, BioSamples from SAMN26555605 to SAMN26555617.

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
