# Peer review of "A Whole Genome Sequencing-Based Epidemiological Investigation of a Pregnancy-Related Invasive Listeriosis Case in Central Italy"

_pathogens, 2022, doi:10.3390/pathogens11060667_

Round 1

Reviewer 1 Report

The manuscript by Valeria Russini et al describes data from a case report and whole genome sequencing-guided investigation of listeriosis.
The manuscript deserves a thorough review before possible publication.

Global:
Original article must be changed to case report.

Values less than 12 should be written in full.
 italicize i.e. ; gene names, ...
prefer passive voice.

Introduction: 
More than an asymptomatic clinical presentation, the French National Reference Center team (Hafner et al. Nat Comm) has recently demonstrated that assymptomatic carriage of Listeria sp. exists in humans and is associated with modulation of the microbiota. This fact needs to be discussed and could provide interesting insight into the situation.

Results: 
The case presentation is very incomplete: when? Why and how was the pregnant woman treated for her listeriosis? The inflammatory indices should be detailed, as well as the CSF parameters.
Paragraph 2.2 : unit are lacking for MIC. why is the AST profile not avalaible for second case. The strain has to be studied, at least by surveillance network. Precise MIC (with unit) must be given for all clinical strains, and not only categorical classification.
Supplementary table 1 must be added in the core manuscript.

Discussion : 
ISS database must be appropriately referenced.

Methods : 
FILMARRAY is not to be capitalized
Why has not MALDI TOF applied on all strains ?
Why have the authors preferend short read sequencing ? 2X300 bp I presume ?
Manufacturer's must be correctly referenced with their name, city and country of location.
Please provide consent for all human cases.

Author Response

Original article must be changed to case report.
We thank the reviewer for the observation. However, a case report is generally a descriptive work, and following the journal guideline, has a different structure that does not mention the “Material and methods”, and focuses on detailed case description. We think that our work offers an original research with the analysis of several data (that involved human, food and environment isolates), the comparisons of different methods to assess outbreaks and epidemiological links, and it is not only a single case presentation. 
In any case, we refer to the opinion of the editor whether to change the structure of our article and change it to a case report.

Values less than 12 should be written in full.
 italicize i.e. ; gene names, ...
prefer passive voice.
Text has been changed according to the reviewer suggestion

Introduction: 
More than an asymptomatic clinical presentation, the French National Reference Center team (Hafner et al. Nat Comm) has recently demonstrated that assymptomatic carriage of Listeria sp. exists in humans and is associated with modulation of the microbiota. This fact needs to be discussed and could provide interesting insight into the situation.
We thank the reviewer for this interesting  suggestion. We added the citation in the main text (introduction and discussion) as the high possibility of asymptomatic carriage of Listeria in the mother. 
Results: 
The case presentation is very incomplete: when?  Why and how was the pregnant woman treated for her listeriosis? The inflammatory indices should be detailed, as well as the CSF parameters.
Thank you for your observation. We have now furtherly detailed the medical history of the presented case. Moreover, we specified the values of PCT and CRP of the newborn. Unfortunately, no lumbar puntcure was performed on the mother, and a wide-spectrum treatment with ampicillin was started due to the documented elevation of CRP and neutrophilic leukocytosis.
Paragraph 2.2 : unit are lacking for MIC. why is the AST profile not avalaible for second case. The strain has to be studied, at least by surveillance network. Precise MIC (with unit) must be given for all clinical strains, and not only categorical classification.
We thank the reviewer for the observation. We add the unit of measurement for MIC, when provided by the hospital. We now performed the AST for all the clinical isolates in our laboratory according to EUCAST. 
Supplementary table 1 must be added in the core manuscript.
The table is now reported in the Results section.
Discussion : 
ISS database must be appropriately referenced.
Text has been changed according to the reviewer suggestion
Methods : 
FILMARRAY is not to be capitalized
Text has been changed according to the reviewer suggestion
Why has not MALDI TOF applied on all strains ?
Unfortunately, not all the institutions (including our laboratory) involved in the investigation are equipped with MALDI TOF. In case of doubtful identification from hospitals (but this was not the case), we proceed with biochemical confirmatory tests and verification of hemolysis according to the ISO 11290-part I 
Why have the authors preferend short read sequencing ? 2X300 bp I presume ?
We thank the reviewer for this question and we now specify this information in main text. Our laboratory is equipped with the ILLUMINA MiSeq instrument and we routinely perform sequencing with 2x300 or 2x250 cartridges depending on the needs. We carry out all the analysis processes in house and we do not rely on any external service for this type of analysis. We also believe that the sequencing method used is suitable for carrying out this type of research and cluster and SNPs analysis.
Manufacturer's must be correctly referenced with their name, city and country of location.
Text has been changed according to the reviewer suggestion
Please provide consent for all human cases.
We thank the reviewer for this observation. As already discussed with the editor, we have the signed consent of the mother of Case A, who is the only case reported in detail in the study. We used the other cases only as case series by a retrospective observational approach and anonymity is guaranteed as the paper does not contain data that allow tracing the identity of the patients (also the name of the town is not specified). We deleted other data of case B, C, D (sex, age, reason for hospitalization) not relevant for the work purpose. The only data we show are the time of bacterial strain isolation and its features (when available). 

Reviewer 2 Report

The manuscript of V. Russini et al. titled "A whole genome sequencing based epidemiological investigation of a pregnancy-related invasive listeriosis case in central Italy" is devoted to revealing hidden outbreak cases of listeriosis in a city near Rome. Although the study and manuscript are well done it has some issues.

The only major issue of the manuscript is the lack of publicly available sequencing data claimed by the authors. The BioProject PRJNA814492 link mentioned in Data Availability Statement has "The following term was not found in BioProject: PRJNA814492". The same is for BioSamples from SAMN26555605 to SAMN26555617. Before submitting a manuscript, authors are required to make the sequencing data publicly available so that anyone can verify the results and conclusions drawn by the authors.

The minor issues and suggestions are below:

Line 37: If all cases were in the same city, could you put the name of the city in Materials and Methods (line 102)?

Lines 45 and 256: Indent a paragraph.

Line 107: Shorten Listeria to L.

Line 172: Omit (TMP/SMX)

Line 207: Replace "gene" with "genes".

Line 231: What is CT?

Figure 3: Change the font or color scheme. Numbers in red cells are almost not readable.

Line 429: Replace with QIAsymphony (Qiagen, Hilden, Germany).

Line 431: Replace (Illumina) with (Illumina, CA, USA).

Line 441: Italisize MLST genes.

Line 454: Version of GrapeTree?

Line 456: Omit "(CGE - www.genomicepidemiology.org)"

Author Response

The manuscript of V. Russini et al. titled "A whole genome sequencing based epidemiological investigation of a pregnancy-related invasive listeriosis case in central Italy" is devoted to revealing hidden outbreak cases of listeriosis in a city near Rome. Although the study and manuscript are well done it has some issues.

The only major issue of the manuscript is the lack of publicly available sequencing data claimed by the authors. The BioProject PRJNA814492 link mentioned in Data Availability Statement has "The following term was not found in BioProject: PRJNA814492". The same is for BioSamples from SAMN26555605 to SAMN26555617. Before submitting a manuscript, authors are required to make the sequencing data publicly available so that anyone can verify the results and conclusions drawn by the authors.
We thank the reviewer for the helpful suggestion. Now we have released the sequences in Genbank.
The minor issues and suggestions are below:

Line 37: If all cases were in the same city, could you put the name of the city in Materials and Methods (line 102)?
We thank the reviewer for the observation. However, we think that the name of the city is an identifying detail and should be omitted to preserve the privacy and anonymity of the cases and the commercial activities involved in the investigation. 
Lines 45 and 256: Indent a paragraph.
Text has been changed according to the reviewer suggestion
Line 107: Shorten Listeria to L.
Text has been changed according to the reviewer suggestion
Line 172: Omit (TMP/SMX)
Text has been changed according to the reviewer suggestion
Line 207: Replace "gene" with "genes".
Text has been changed according to the reviewer suggestion
Line 231: What is CT?
Core genome MLST type. Text has been changed.
Figure 3: Change the font or color scheme. Numbers in red cells are almost not readable.
We changed the font colour in the figure and now it should be more readable.
Line 429: Replace with QIAsymphony (Qiagen, Hilden, Germany).
Text has been changed according to the reviewer suggestion
Line 431: Replace (Illumina) with (Illumina, CA, USA).
Text has been changed according to the reviewer suggestion
Line 441: Italisize MLST genes.
Text has been changed according to the reviewer suggestion
Line 454: Version of GrapeTree?
The version has been added
Line 456: Omit "(CGE - www.genomicepidemiology.org)"
Text has been changed according to the reviewer suggestion

Reviewer 3 Report

Russini et al. describe the application of whole genome sequencing to investigation of listeriosis outbreak in Italy. In general, the manuscript is well written and easy to comprehend, and the study design is appropriate. The data analysis was thorough and scientifically correct. Successful application of WGS to outbreak investigation will definitely attract an increased attention of potential manuscript readers. I have only several minor comments, mostly representing typos, to be addressed.

Minor comments:

Line 64“clinical features … includes” – should be “include”

Line 102 – please add year to the date (2020?)

Line 103 – should be either “31st week” or “31 weeks”

Line 109 – a number should be in uppercase in ‘mm3’

Line 109 – does 33.000 stand for thirty three thousand? Then it should be written as “33,000” or “33 000”

Line 152 – “see above” – probably, should be “see below” since the previous chapters did not provide serotype info

Line 160 – there should be dash or slash sign between trimethoprim and sulfamethoxazole

Line 199 – “in silico” should be italicized, not “analysis”

Line 200 – “three human cases (A, B, C, D)” – please fix

Line 256 – according to the work by Schurch et al. (https://doi.org/10.1016/j.cmi.2017.12.016) the cgMLST allele difference threshold for L. monocytogenes could be set to 3, so this confirms the assignment of the isolates to two clusters. However, wgMLST cutoff in this work was set to 10. I do agree that the threshold value is highly debatable and could depend on outbreak conditions. At the same time, I think that the manuscript in general and cgMLST tree in particular would benefit from addition of outgroup isolate of the same ST (e.g., L. monocytogenes from other region or country), which would help to justify the threshold value for this outbreak investigation.

Line 303 – “Since this evidence” – does not make sense, please rephrase

Line 315 – should be “cases B and C”

Line 341 – “it would have been useful further investigations” – a verb is missing (to perform further…?)

Line 441 – gene name are usually italicized

- please fix “corrispondig” in Supplementary table

Author Response

Russini et al. describe the application of whole genome sequencing to investigation of listeriosis outbreak in Italy. In general, the manuscript is well written and easy to comprehend, and the study design is appropriate. The data analysis was thorough and scientifically correct. Successful application of WGS to outbreak investigation will definitely attract an increased attention of potential manuscript readers. I have only several minor comments, mostly representing typos, to be addressed.

Minor comments:

Line 64 – “clinical features … includes” – should be “include”
Text has been changed according to the reviewer suggestion
Line 102 – please add year to the date (2020?)
Text has been changed according to the reviewer suggestion
Line 103 – should be either “31st week” or “31 weeks”
Text has been changed according to the reviewer suggestion
Line 109 – a number should be in uppercase in ‘mm3’
Text has been changed according to the reviewer suggestion
Line 109 – does 33.000 stand for thirty three thousand? Then it should be written as “33,000” or “33 000”
Text has been changed according to the reviewer suggestion
Line 152 – “see above” – probably, should be “see below” since the previous chapters did not provide serotype info
Text has been changed according to the reviewer suggestion 
Line 160 – there should be dash or slash sign between trimethoprim and sulfamethoxazole
Text has been changed according to the reviewer suggestion
Line 199 – “in silico” should be italicized, not “analysis”
Text has been changed according to the reviewer suggestion
Line 200 – “three human cases (A, B, C, D)” – please fix
Text has been changed according to the reviewer suggestion
Line 256 – according to the work by Schurch et al. (https://doi.org/10.1016/j.cmi.2017.12.016) the cgMLST allele difference threshold for L. monocytogenes could be set to 3, so this confirms the assignment of the isolates to two clusters. However, wgMLST cutoff in this work was set to 10. I do agree that the threshold value is highly debatable and could depend on outbreak conditions. At the same time, I think that the manuscript in general and cgMLST tree in particular would benefit from addition of outgroup isolate of the same ST (e.g., L. monocytogenes from other region or country), which would help to justify the threshold value for this outbreak investigation.
We thank the reviewer for this helpful suggestion. For the assessment of the threshold of allelic distances, we used the guidelines of JOINT ECDC–EFSA for RAPID OUTBREAK ASSESSMENT https://www.ecdc.europa.eu/sites/default/files/documents/listeria-multi-country-out se definition (7 allelic distance in cgMLST). In addition in this guidelines https://doi.org/10.2903/j.efsa.2019.5898 indicate the paper by Moura et al., 2016 as referral point for the analysis of cgMLST and distance outbreak definition.  break-october-2018.pdf that point out the laboratory criterion for outbreak case definition (7 allelic distance in cgMLST). In addition in this guidelines https://doi.org/10.2903/j.efsa.2019.5898 indicate the paper by Moura et al., 2016 as referral point for the analysis of cgMLST and distance outbreak definition. 

We added  in the Supplementary section the analysis of the cgMLST with other food samples processed in our laboratory belonging to ST5 as outgroup of the outbreak and the result section has been changed accordingly. 
Line 303 – “Since this evidence” – does not make sense, please rephrase
Text has been changed according to the reviewer suggestion
 Line 315 – should be “cases B and C”
Text has been changed according to the reviewer suggestion
Line 341 – “it would have been useful further investigations” – a verb is missing (to perform further…?)
Text has been changed according to the reviewer suggestion
Line 441 – gene name are usually italicized
Text has been changed according to the reviewer suggestion
- please fix “corrispondig” in Supplementary table
Text has been changed according to the reviewer suggestion

Round 2

Reviewer 1 Report

The manuscript has been revised according to my previous comments.

Reviewer 2 Report

The authors have done a thorough work on the issues and the manuscript can be accepted for publication.